# Accuracy of Tracking Devices’ Ability to Assess Exercise Energy Expenditure in Professional Female Soccer Players: Implications for Quantifying Energy Availability

**DOI:** 10.3390/ijerph19084770

**Published:** 2022-04-14

**Authors:** Marcus S. Dasa, Oddgeir Friborg, Morten Kristoffersen, Gunn Pettersen, Jorunn Sundgot-Borgen, Jan H. Rosenvinge

**Affiliations:** 1Department of Health and Care Sciences, UiT, The Arctic University of Norway, 9019 Tromso, Norway; gunn.pettersen@uit.no; 2Department of Psychology, UiT, The Arctic University of Norway, 9019 Tromso, Norway; oddgeir.friborg@uit.no (O.F.); jan.rosenvinge@uit.no (J.H.R.); 3Department of Sport, Food and Natural Sciences, Western Norway University of Applied Sciences, 5063 Bergen, Norway; morten.kristoffersen@hvl.no; 4Department of Sports Medicine, Norwegian School of Sport Sciences, 0863 Oslo, Norway; jorunnsb@nih.no

**Keywords:** female athlete, exercise expenditure, energy availability, team sport, exercise metabolism, technology

## Abstract

The purpose of the study was to assess the accuracy of commonly used GPS/accelerometer-based tracking devices in the estimation of exercise energy expenditure (EEE) during high-intensity intermittent exercise. A total of 13 female soccer players competing at the highest level in Norway (age 20.5 ± 4.3 years; height 168.4 ± 5.1 cm; weight 64.1 ± 5.3 kg; fat free mass 49.7 ± 4.2 kg) completed a single visit test protocol on an artificial grass surface. The test course consisted of walking, jogging, high-speed running, and sprinting, mimicking the physical requirements in soccer. Three commonly used tracking devices were compared against indirect calorimetry as the criterion measure to determine their accuracy in estimating the total energy expenditure. The anaerobic energy consumption (i.e., excess post-exercise oxygen consumption, EPOC) and resting time were examined as adjustment factors possibly improving accuracy. All three devices significantly underestimated the total energy consumption, as compared to the criterion measure (*p* = 0.022, *p* = 0.002, *p* = 0.017; absolute ICC = 0.39, 0.24 and 0.30, respectively), and showed a systematic pattern with increasing underestimation for higher energy consumption. Excluding EPOC from EEE reduced the bias substantially (all *p*’s becoming non-significant; absolute ICC = 0.49, 0.54 and 0.49, respectively); however, bias was still present for all tracking devices. All GPS trackers were biased by showing a general tendency to underestimate the exercise energy consumption during high intensity intermittent exercising, which in addition showed a systematic pattern by over- or underestimation during lower or higher exercising intensity. Adjusting for EPOC reduced the bias and provided a more acceptable accuracy. For a more correct EEE estimation further calibration of these devices by the manufacturers is strongly advised by possibly addressing biases caused by EPOC.

## 1. Introduction

To support basic physiological functions and aid adaptations to training, an athlete’s energy intake (EI) should be matched against the energetic needs a given sport activity require. An athlete’s energy availability (EA) is quantified as the residual energy after subtracting the exercise energy expenditure (EEE) from the EI, divided by fat-free mass (FFM) [1]. In soccer, nutritional intake may impact a player’s body composition, resulting in performance alterations. As such, sport nutrition experts can assist players in manipulating EI to meet the desired goals [2]. Nevertheless, athletes should be warned against the accidental or deliberate mismatch of EI and energy expenditure (EE), resulting in EA below 125 kilojoules (kJ) (30 kcal) per kg FFM. Such a low energy availability (LEA) may cause disturbances to hormonal, metabolic, and immune functions [3,4,5]. Although monitoring of body mass can provide insight into an athlete’ s energy balance, long term LEA may result in “metabolic adaptation” causing weight stability, despite inadequate energy balance [6]. Thus, body mass is not sufficient to detect LEA in athletes. Despite this, limited knowledge exists regarding the occurrence and implications of LEA in female soccer players.

In soccer and intermittent field sports, tracking devices (GPS or accelerometer-based) is the most commonly used microtechnology to quantify physical activity [7], and to estimate EA [8,9,10]. Such devices may provide valid estimates of EEE within sports characterized by steady-state exercise such as running and cycling [11]. However, their accuracy is questionable within sports characterized by intermittent exercises, notably soccer, with high amounts of directional changes, accelerations, and deaccelerations [12]. In part, this is due to the anaerobic energy production, resulting in lactate accumulation and oxidation, which is not accounted for in aerobic energy production and therefore difficult to measure [13]. Further, different manufacturers may operate with disparate algorithms when processing data, potentially resulting in dissimilar output of EEE [14,15].

Some tracking devices, which are specifically developed for intermittent sports such as soccer, build on the metabolic power concept [16,17], yet several studies [12,18,19,20] still report underestimations of EEE. Responding to such findings, the original authors have argued that inappropriate usage of the concept could contribute to this underestimation [21]. Others have proposed alternatives to the original model [22]. Nevertheless, the validity of studies reporting EEE in intermittent sports based on tracking devices is ambiguous. This needs to be further investigated in female athletes and soccer players, as previous work on metabolic power is based on male athletes. Disparities in physiological factors between sexes such as work economy, efficiency, and body composition are also contradictory [23] and have not been addressed in the development of the metabolic power concept.

In a recent study [24] of female endurance athletes, only a slight caloric surplus of <200 kcal·d^−1^ in energy balance was associated with increased performance. However, failure to achieve a caloric surplus was associated with impairments to performance. This finding highlights the importance of accurately measuring EEE, as it may have direct consequences for the nutritional periodization of athletes, influencing performance, recovery, and health status. In summary, there is a need to identify the accuracy of current tracking devices being utilized to quantify EEE, offering implications for measuring LEA. Further, the investigation of female athletes is warranted to examine potential differences in estimating EEE and enable between study comparisons, regardless of sex.

Therefore, the aim of the present study was to examine the accuracy of three commonly used tracking devices utilizing metabolic power to quantify EEE during intermittent exercise in high-level female soccer players. Based on the current literature, we expected that the tracking devices underestimate the caloric expenditure as compared with indirect calorimetry as the criterion measure. we also examined whether EPOC during rest could be used to improve agreement and explain potential discrepancies in the results.

## 2. Materials and Methods

### 2.1. Study Design

Participants completed a single visit test protocol on artificial grass surface instrumented with a portable O_2_ analyzer, and three different tracking devices. A pre-determined course consisting of walking, jogging, shuttle run/stride, and sprinting was designed to model the physical requirements in women’s soccer [25,26]. The course length was 549.5 m and was repeated five times (total distance 2747.5 m) to ensure data sufficiency to measure movement and EEE (Figure 1). Participants were instructed to complete each part of the course at self-selected speeds, guided by movement descriptors. However, the 20 m sprint was instructed to be completed at maximal effort. Before starting the testing protocol, participants were instructed in how to use the Rated Perceived Exertion (RPE) [27] scale ranging from 0–10. All participants completed a standardized guided warm up, consisting of three rounds, corresponding to RPE 4, 6 and 8, respectively, the latter being the desired intensity for the completion of the 5-round protocol (self-selected speed corresponding to RPE 8). After each round, they rested standstill for 1 min. Here capillary blood lactate samples were collected, and the participants were asked to rate their RPE of the preceding round. Additionally, after the completion of the protocol, excess post exercise oxygen consumption (EPOC) was measured for 120 s to account for EEE derived work above VO_2_ max and total session RPE was stated.

### 2.2. Participants

Eligibility criteria for the study were defined as (i) female competing at the highest level in Norway, (ii) >16 years of age, and (iii) absence of injuries or illnesses. In addition, participants were asked to abstain from caffeine intake on the day of testing, as well as ingesting their last meal approximately 2 hours before testing. A total of 13 professional female soccer players (age 20.5 ± 4.3 years; height 168.4 ± 5.1 cm; weight 64.1 ± 5.3 kg; fat free mass 49.7 ± 4.2 kg) completed the study. Two players declined the invitation to participate due to self-reported injury and time commitment. Following the Helsinki declaration, all participants were informed about the project both orally and in writing and signed an informed consent document. The project was approved by the Norwegian Center for Research Data (Reference: 807592).

### 2.3. Tracking Measures

Participants were equipped with an 18 Hz GPS device with 952 Hz tri-axel accelerometer, gyroscope, and magnetometer (GPS^1^, Apex, StatSport, Newry, Northern Ireland, UK), a 10 Hz GPS device with 1 kHz tri-axel accelerometer, gyroscope and magnetometer (GPS^2^, Vector, Catapult innovations, Melbourne, Australia), and a 1000 Hz inertial sensor device, with accelerometer, gyroscope and multi-chip motion tracking module (IMU, Playermaker^TM^, Tel Aviv, Israel). All devices were mounted and used according to manufacturers’ guidelines. The GPS devices were securely positioned in a custom-made vest 2–3 cm apart, between the participants’ scapulae. Both GPS devices were placed outside in record mode at least 20 min prior to testing, to ensure adequate satellite connection. The inertial sensor was mounted on the participants boots, using the manufacturers boot strap designed for this purpose. After completion, data were uploaded to the device-specific software and analyzed to calculate EEE for the whole period, before being exported to Microsoft Excel. After reaching out to the manufacturers of GPS^1^ and GPS^2^, both confirmed that the calculation of metabolic power builds on previous work by Osgnach et al. [16], utilizing acceleration and velocity data for the calculation. For the inertial measuring device (IMU), EEE calculations were done by the manufacturer as the software lacked this feature. The technical properties of this device is explained elsewhere [28] and it applies the same metabolic power method [16]. Specifically, speed and acceleration were calculated in 10 Hz, together with the formula and constants provided in the algorithm by [16]. All tracking data were also edited post hoc, by synchronizing the start and cessation of the protocol with the oxygen consumption (VO_2_)-derived data, ensuring the same measurement time for the various devices.

### 2.4. Indirect Calorimetry

Indirect calorimetry (VO_2_ Master Health Sensors Inc., Vernon, BC, Canada) was used to establish VO_2_-derived EEE and served as criterion measure against the tracking systems. The VO_2_ master have previously been validated [29] resulting in a difference ranging from 0.17–0.27 VO_2_ (L/min) during different intensities, compared to the Parvomedics trueOne 2400 metabolic cart (Parvomedics, Inc., Salt Lake City, UT, USA). Participants wore the VO_2_ master for the entire protocol, including rest periods, as well as 120 s following the last round to account for EPOC, following the intermittent exercise protocol. To establish resting energy expenditure, participants wore the VO_2_ master for 10 min, after arriving at the facility, laying down in supine position. The mean VO_2_ (L/min)-derived EE value from the last 5 minutes was subtracted from the total EE during the test post hoc for each individual, consistent with previous literature [20,30]. The VO_2_ master was calibrated according to the manufacturer’s guidelines prior to each testing session. After completion, breath by breath analysis of VO_2_ (L/min) was analyzed in 30 s intervals, before being converted to kJ to establish VO_2_ derived EEE using a respiratory exchange ratio (RER) of 1.00, indicating mainly glycolytic energy production [31]. This was done as the usage of RER assumes constant oxygen content and that CO_2_ exchange in the lungs reflects that of the cells [32]. As this is not the case during intermittent exercise, lactate measurements served as confirmation of the appropriate RER level chosen.

### 2.5. Lactate Measurement

Blood lactate (mmol/L) was measured using the lactate plus (Lactate Plus, Nova Biomedical, Waltham, MA, USA), which have previously been validated [33]. Samples were taken at rest from the index finger, following resting energy expenditure measurement and after each round of the protocol, indicating the level of intensity for the completed work. Thus, blood lactate measures were used to verify the RER used to calculate EEE, as blood lactate is associated with substrate metabolism during exercise [34].

### 2.6. Statistical Analyses

The statistical analyses and preparation were conducted using SPSS 26 (IBM, Armonk, NY, USA) and Microsoft Excel (Microsoft corporation, Redmond, WA, USA). Descriptive statistics from the participants are given for total running distance, inter-device distance, percentage difference, as well as the lactate levels for the separate circuit rounds. All devices were directly compared to the criterion measure (VO_2_ derived EEE) and intraclass correlation coefficients (ICC) were estimated to determine the level of agreement between the individual tracking devices and the criterion measure. We estimated two-way mixed ICC models with the subjects and method factors as random and fixed, respectively [35]. ICC estimates are presented based on the single measure formula since a single device score represents the EEE score. ICC estimates for both relative (consistency) and absolute agreement are given along with their 95% confidence intervals.

Paired sample t-tests were used to examine for mean differences between the measurement methods, thus indicating the general level of bias. Effect sizes (ES) for these differences were calculated by dividing on the standard deviation of the difference scores corrected for their correlation. We report Hedge’s *g*, which additionally corrects bias related to smaller samples, thus reducing overestimation according to the formula [36]: $g=\frac{\bar{M_{1}-M_{2}}}{\sqrt{\left(s_{1}^{2}+s_{2}^{2}-2rs_{1}s_{2}\right)}\;/\;\sqrt{2\left(1-r\right)}}\times J$ (with *J* as the Hedge’s *g* correction factor according to [37]. The a priori alpha level was set to *p* < 0.05. We added linear regression analyses to examine the level and direction of systematic biases between the methods. The tracking device in question was used as a predictor with VO_2_ as the outcome. Unstandardized residuals, which represent the difference between the predicted and the actual VO_2_ energy consumption score was saved and plotted on the y-axis against VO_2_ data on the x-axis. Systematic biases would be present if the residual scores showed a non-flat increasing or decreasing pattern depending on the actual VO_2_ levels. Lastly, we examined if adjusting the EEE calorimetry scores by subtracting the resting time or the EPOC scores could reduce bias and yield better agreement according to new paired sample *t*-tests, smaller residual scores and improved ICC estimates. All results are presented as mean ± SD, unless specified.

## 3. Results

The total distance and mean energy expenditure measured for the tracking devices GPS^1^, GPS^2^, and IMU are presented in Table 1. Compared to the manually measured track, results for distance displayed a percentage difference of 4.4%, 3.7%, and 0.7%, respectively. Mean lactate measurement was significantly elevated compared to baseline resting values (1.3 ± 0.4 mmol/L) during round 1–5 (Figure 2).

EEE measured as indirect calorimetry (criterion measure) compared to EEE estimated by GPS^1^, GPS^2^, and IMU (EEE^IMU^) is presented in Table 1. Compared to the indirect calorimetry, all tracking devices significantly underestimated the caloric expenditure during intermittent bouts of exercise (GPS^1^, *p* = 0.022, ES = 0.60, GPS^2^, *p* = 0.002, ES = 0.96 and IMU, *p* = 0.017, ES = 0.77). When adjusting EEE by subtracting EPOC (EEE measured during standstill resting periods) from the measurement, no differences were found (EEEGPS^1^
*p* > 0.05, ES = 0.44, EEE^GPS2^
*p* > 0.05, ES = 0.15 and EEE^IMU^
*p* > 0.05, ES = 0.21) (Table 1).

The ICC values for total EEE ranged between 0.48 and 0.21 based on consistency estimation, and between 0.39 and 0.24 based on absolute agreement estimation, respectively. Adjusting calculations by excluding EPOC measurements, only analyzing moving time, ICC values ranged between 0.54 and 0.48 and between 0.54 and 0.49 based on consistency and absolute agreement estimation, respectively (See Table 1 for specific values).

Series of regression analyses with VO_2_ as outcome and the specific tracking device as predictors are given in Table 2. An unstandardized beta coefficient above or below 1 indicates under- versus overestimation, respectively. Using mean centered values as predictors, the unstandardized coefficients for EEE^GPS1^ were 1.42. The GPS^1^ device thus underestimated true calorimetry usage with 0.42 kJ per unit increase in the VO_2_ measurement. For EEE^GPS2^, the unstandardized coefficient was 1.82; hence, underestimating estimated caloric expenditure with a mean of 0.82 kJ, compared to the criterion measure. Lastly for EEE^IMU^, unstandardized coefficient was 1.28, yielding a mean underestimation of 0.28 kJ.

Adjusting the caloric estimation by removing EPOC from the estimated EEE_VO2_ score, produced lower unstandardized coefficients, hence yielding lower mean differences between estimated and true caloric expenditure values (0.23 kJ for GPS^1^, 0.66 kJ for GPS^2^ and 0.11 kJ for IMU respectively (Table 2). The regression analysis also displays increased disagreement between predicted EEE values for tracking devices, compared to the observed EEE, as caloric expenditure increases. Thus, increased bias is expected when the caloric expenditure increases during intermittent activity (Figure 3).

## 4. Discussion

This is the first study to quantify the accuracy of the latest tracking devices in female athletes. Our results show that all tracking devices underestimated the EEE and, thus, failing to adequately estimate caloric expenditure, yet GPS^1^ provided the most accurate results. The level of underestimation shrunk for all devices when the criterion measure (indirect calorimetry) was adjusted by subtracting EPOC (the standstill rest periods) from the estimation of energy expenditure. Furthermore, the tracking devices displayed a systematic pattern of bias by overestimating EEE at lower levels of caloric expenditure and underestimating EEE at higher levels of caloric expenditure.

These findings align with several previous studies conducted among male athletes, reporting that GPS and accelerometer-based tracking devices fail to accurately estimate total EEE in intermittent team sports [12,18,20]. Albeit with older technology, Brown et al. [16] reported that GPS units displayed reasonable accuracy during steady-state jogging and running [20]. However, substantial underestimation was observed during intermittent movements and high-intensity actions. Stevens et al. [16] reported that metabolic power [16] overestimated EEE during continuous, steady-state running. Conversely, these authors also found that metabolic power underestimated EEE during aerobic shuttle running. As such, it is possible that the metabolic power algorithm systematically overestimate EEE during walking and jogging and underestimate during high-intensity circuit movements, resulting in total underestimation of EEE during high-intensity intermittent exercise. Recently, Savoia et al. [16] proposed an alternative metabolic power algorithm based on the original work [22]. Here the authors claim that previous studies demonstrating underestimation of EEE using metabolic power, utilize prolonged periods of rest, not reflecting the actual demands of the game. This may be of importance in the interpretation of the present, and the previous studies, as metabolic power primarily relies on locomotion when estimating EEE. Nevertheless, our results demonstrate that three of the latest and most-used tracking devices in modern soccer all underestimate EEE with its current technology.

In the present study, all three devices displayed total distances within 4.4% of the manually measured track. However, the metabolic power equation bases its calculations on velocity and accelerations; thus, the inability of tracking devices to accurately estimate this will influence the total estimated EEE. Previous research has found interindividual differences between devices utilizing metabolic power, although displacement measures were relatively similar, indicating disparities in the filtering of the GPS data [15]. Nonetheless, acceptable validity and reliability for GPS devices of 10 and 18 Hz, as used in this study, have been reported [38]. Further, the inertial sensor used have been compared against high sampling GPS units [28]. Although SD varied between the specific devices, this alone is unlikely to explain the discrepancy in EEE. Hence, the underestimation seen during intermittent exercise might be highly influenced by the algorithm applied by the devices, rather than inaccurate sampling rates or measurements of velocity/acceleration. This assumption is strengthened when investigating the results with and without EPOC measures. Several assumptions are made in the modeling of metabolic power, especially during high intensity running, including running efficiency [17,39]. Further, surface may also play an influential role on the energetic cost of running [40], together with individual running economy. As the metabolic power model is based on well-trained male endurance athletes and this study was done in female soccer players on an artificial surface, these factors may well be partially responsible for some of the observed discrepancies. Future studies could therefore consider tailoring the metabolic power equation for females. Further, practitioners may consider applying individual data for running cost to the equation responsible for the EEE output. This would require substantial testing of each athlete, as well as post session/match editing of the GPS derived data to calculate the EEE using the athlete specific data as constants in the metabolic power equation. Since individuals clearly differ in response to estimates of metabolic power based on average data, this could be of interest in athletes where accurate measurement of energy availability is of special importance for health and performance outcomes.

Our results show an inverse relationship as EEE^VO2^ increases. In addition, individual lactate measurements increase together with EEE^VO2^. As such, it appears that the metabolic power estimates are somewhat correlated with exercise intensity. This is confirmed when adjusting the analysis by subtracting EPOC measurements, resulting in non-statistically significant differences between all devices and EEE^VO2^. These results indicates that tracking devices are unable to sufficiently account for anerobic energy metabolism, manifested by elevated VO_2_ levels during rest (e.g., replenishing substrate stores, repaying O_2_ debt from the previous high-intensity action) [13], similar to previous research [18].

Several studies investigating LEA in female athletes have used devices relying on metabolic estimates for EEE, to calculate EA [9,10,41,42]. Nonetheless, based on the findings of our study, caution should be taken when interpreting results from studies utilizing algorithmical estimates to quantify EEE. Undoubtedly, the main challenge when identifying LEA in athletes is the definition of EA itself, as it relies heavily on measures of EI and EEE, both fragile for significant error [11]. As such, more objective physiological markers have been proposed going forward within the field of LEA [43]. These include the usage of hormonal data; however, more research is needed regarding the sensitivity and specificity of distinctive markers.

The sample size may raise concerns with response to statistical power. This could have been increased with repeated measures, which was not possible due to the heavy schedule of the players. Nevertheless, anthropometric characteristics are similar to those reported in international and high domestic leagues elsewhere [41]. As such, we will argue that the present results are applicable to other elite female soccer players. Moreover, power is not critical as the main aim of the study was not to test specific hypotheses, but to test the level of accuracy against indirect calorimetry for each tracking device. Conversely, power is a concern in the sense that increased power would have allowed us to test accuracy across the different tracking devices. We chose extended periods of rest between rounds and post activity, to account for EPOC. During actual gameplay, prolonged periods of total rest are sparser and, thus, may have contributed to the deviation between tracking devices and VO_2_ measurements. We also acknowledge that 120 s of EPOC measurement post-test is not sufficient to return to baseline levels, hence, expanding the time would likely increase discrepancy. However, given the slope of EPOC [42], 120 s will encompass the majority of significant elevation in energy expenditure.

## 5. Conclusions

To our knowledge, this is the first all-female study exploring estimated EEE, as well as comparing the latest model of three widely used tracking devices. The GPS and accelerometer-based tracking devices tested generally underestimate the caloric expenditure during intermittent exercise in professional female soccer players. This is primarily because such devices cannot account for anaerobic energy production seen during high-intensity exercise. Furthermore, the observed differences between manufacturers could be of importance for practitioners and their choice of equipment. Therefore, caution should be taken when utilizing estimated EEE in calculations of EA and nutritional calculations. This is of special importance in training situations, where increased rest times between drills and play is likely to produce greater underestimation of total EEE. Nevertheless, as calculations based on EEE is the only method for assessing players EA at this point, the usage of devices applying metabolic power is presumably superior to standard GPS and heart rate measures. However, the deviations seen in caloric expenditure must be considered by practitioners and researchers depending on the need of accuracy. Future studies should also aim to include female players in the validation of the algorithms, as well as individualizing the algorithm. Despite generally underestimating EEE, the devices tested can still provide useful information in quantifying EA, taking the highlighted limitations in consideration.

## Figures and Tables

**Figure 1 ijerph-19-04770-f001:**
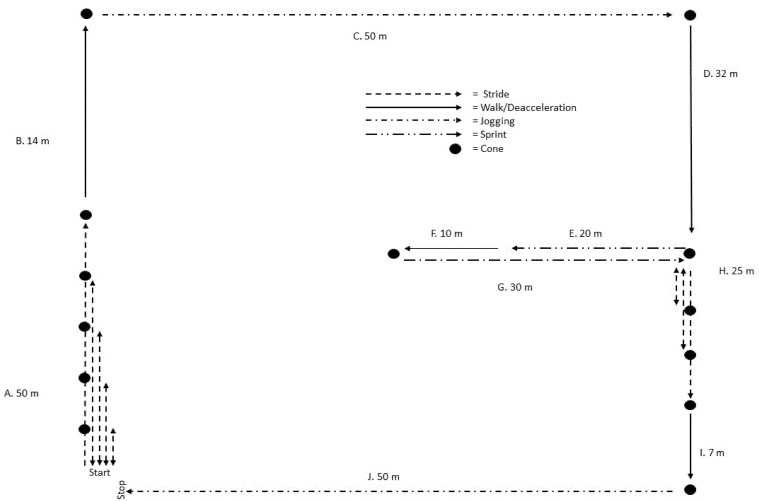
Illustration of the intermittent exercise protocol, indicating the type of movement and length of segment numbered A–J.

**Figure 2 ijerph-19-04770-f002:**
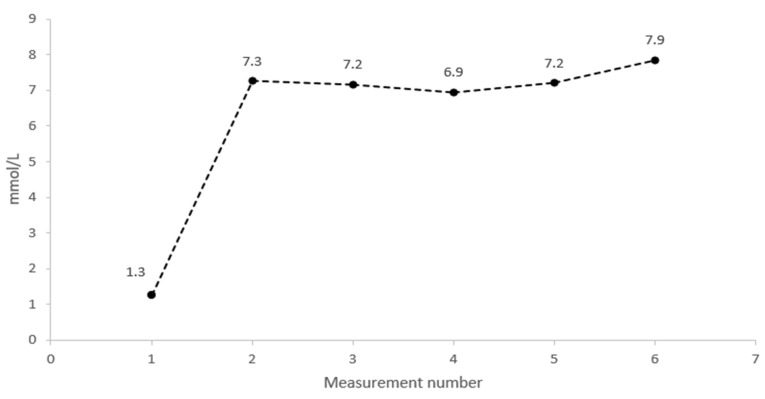
Time plot of the lactate measurement levels during the protocol, at baseline (rest) and the following each completed round.

**Figure 3 ijerph-19-04770-f003:**
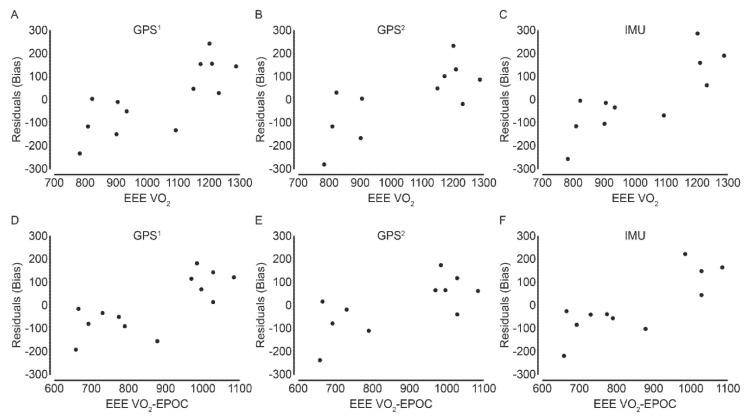
Residual plot indicating the disagreement between the predicted EEE (residuals) and measured EEE_VO2_ in the upper panel and EEE_VO2_-EPOC in the bottom panel (values displayed in kJ). Negative values indicate overestimation and positive values indicate underestimation. Exercise energy expenditure = EEE; kilojoules = kJ; oxygen consumption = VO_2_; excess post-exercise energy consumption = EPOC.

**Table 1 ijerph-19-04770-t001:** Upper panel displays descriptive data for the criterion measure EEE^VO2^, distance measured by devices, and total EEE for the individual tracking devices. Middle panel display ICC measures for absolute and consistency measures, percentage error (EEE), as well as paired sample *t*-test between specific devices and criterion measure for total energy expenditure. Lower panel of the table display the values adjusted, by removing EPOC measurement from the criterion measure value (VO_2_), only analyzing moving time. Exercise energy expenditure = EEE; kilojoule = kJ; effect size = ES; post-exercise energy consumption = EPOC.

	GPS^1^	GPS^2^	IMU
N	13	11	11
VO_2_^EEE^ (kJ)	1038 ± 183	1043 ± 198	1016 ± 191
Distance (% error)	2625 ± 25 (4.4%)	2644 ± 73 (3.7%)	2767 ± 207 (0.7%)
EEE (kJ)	933 ± 83	843 ± 73	879 ± 82
ICC^ABS^	0.39	0.24	0.30
ICC^CON^	0.48	0.44	0.42
Percentage error	10.7%	20.6%	14.5%
*p* value	0.022	0.002	0.017
ES	0.60	0.96	0.77
Values adjusted for EPOC			
VO_2_^-EPOC^ (kJ)	868 ± 156	875 ± 168	847 ± 161
ICC^ABS^	0.49	0.54	0.49
ICC^CON^	0.54	0.53	0.48
Percentage error	7.2%	3.1%	3.7%
*p* value	>0.05	>0.05	>0.05
ES	0.44	0.15	0.21

**Table 2 ijerph-19-04770-t002:** Display regression coefficients representing the mean change for specific devices as predicted by the regression model, compared against the criterion measure VO_2_. The first part of the table displays the values for total EEE, with the second part showing values adjusted by removing EPOC measurements, only analyzing moving time.

	GPS^1^	GPS^2^	IMU
N	13	11	11
Intercept	1038.5	1043.1	1016.2
Beta	1.42	1.82	1.28
t	2.75	2.75	2.1
Absolute residual error (kJ)	112.3 ± 78.7	109.1 ± 89.4	121.6 ± 92.3
*p* value	0.019	0.022	0.077
95% CI	0.3–2.5	0.3–3.3	0–2.6
Values adjusted for EPOC are presented below			
Intercept	867.8	875.4	846.8
Beta	1.23	1.66	1.11
T	2.86	3.16	2.1
Absolute residual error (kJ)	97.3 ± 59.9	89.1 ± 67.2	103.6 ± 72.1
*p* value	0.015	0.011	0.69
95% CI	0.3–2.2	0.5–2.8	0–2.2

## Data Availability

Data presented are available on request from the corresponding author.

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
