# Peer review of "Accuracy of Tracking Devices’ Ability to Assess Exercise Energy Expenditure in Professional Female Soccer Players: Implications for Quantifying Energy Availability"

_ijerph, 2022, doi:10.3390/ijerph19084770_

Round 1

Reviewer 1 Report

IJERPH-1628875

The manuscript entitled “Accuracy of Microtechnology to Assess Exercise Energy Expenditure in Professional Female Soccer Players:  Implications for Quantifying Energy Availability” aim to assess the accuracy of commonly used GPS/accelerometer-based tracking devices in the estimation of exercise energy expenditure (EEE) during high-intensity intermittent exercise.  

The study addresses an important topic in the developing technology field to measure fitness and related outcomes.

 Yet, the sample is very small (13 individuals), and the large age range (indicating an inhomogeneous sample) raises doubts regarding the reliability of estimates. Please discuss these limitations in more detail.

Although the authors said in the discussion that “the sample size is 318 not deemed critical for the interpretation of the present findings”, this should be supported by a sample power analysis. 

Please acknowledge the limitations of this study in the discussion section. 

 Please detail the conceptual advancements in more depth.

 What would you recommend to coaches as well as manufacturers?

Author Response

Please see the attached PDF file for in-depth response to the comments made 

Reviewer 2 Report

This is a very interesting and important manuscript. However, some points shoud be adressed:

the athletes repeated the training course for 5 times. Between the lapses they stood still for 1 minute getting the lactate analysis. During these phases EPOC will sum up. So why do you calculate EPOC for 120 seconds after completing the experiment. Please explain.

Are there any estimations to what extent the experimental physical load during the experimental circuit corresponds to the actual performance in the competition? What about the EPOC during a soccer game, are there any data available to calculate or estimate EPOC during a soccer game? Once the authors try to enhance accuracy of exercise energy expenditure, they should derive recommendations from their data to make in-game measurements with these microtechnological devices more accurate. The three used devices have different features and sampling frequencies. Could the authors discuss, what kind of technique may be superior to another? Is the frequency crucial (are the data in this respect at all meaningful enough? If not, please mention this as a limitation). Furthermore, the primary endpoint was to examine the accuracy of three devices. How was the required number of subjects determined, could you please refer to a sample size calculation? And if you want to examine accuracy, don`t you also have to do repeated measurements, so you know, that the device leads to stable results? Please discuss this point in the discussion.

line 141 please correct the sentence

Author Response

Please see the attachment for in-depth response to the comments made

Reviewer 3 Report

General

This manuscript reports the results of a comparison of three tracking devices to a gold standard, indirect calorimetry, in female soccer players. The results show rather poor accuracy and also poor precision, which slightly improve when taking into account EPOC.

Major

I was not surprised, that, again, such tracking devices were shown to suffer from bias as compared to some reasonable standard. Since the authors rational was to (in)validate the use of such tracking devices for energy availability purposes, it would seem to me that the attention should not be on some averages but rather on what it means for an individual. Say that a new adapted algorithm would yield a perfect on average match, but likely still with an SD, this would potentially still lead to individual athletes being wrongly advised to increase or decrease their energy intake. So wouldn’t it be better to calibrate such devices for each individual? Using the soccer-like circuit, indirect calorimetry and tracking device would be collected in parallel during a first session, and then the algorithm is adapted to get a perfect match for the individual, and then a validation circuit can be performed to verify the accuracy of the individually adapted algorithm. I know that this was not done in this study, but perhaps the authors could discuss such a way for the future? Their argument in the discussion that physiology is similar between individuals is rather flawed I find.

The idea to take into account EPOC was sensible, and did indeed improve somewhat the accuracy of the three devices in comparison with indirect calorimetry. However, why measure EPOC only during two minutes? Given the lactate levels reached (8 mmol/L) 2 minutes was likely not enough to get back to baseline. It also would have been good to have some more lactate points to show the rate of decrease. In other words, some EPOC was probably missed, and if it had been taken into account could have improved even more the comparison. This should be mentioned in the limitations section.

I am not a statistician and therefore have limited authority with regard to the ways the data were analyzed. However, I wondered why the authors did not use Bland-Altman plots to illustrate the bias between the devices and the gold standard. The advantage of Bland-Altman plots is that one sees immediately how well a measure does agains some standard. If there is a good reason to not do Bland-Altman plots please explain, otherwise I would suggest to show those plots.

The results of the linear regression analysis are only given numerically, which can be very misleading, since similar results can be obtained with different distributions. I would like to see the individual data points and the regression lines and an extra figure.

Finally, and please do not take this personally, but I find the writing somewhat wooly and convoluted, especially in the discussion. Some effort in making the argument in a more concise way would help improve readability.

Minor

Throughout, there are still quite some typos, syntax errors, missing words, and a thorough editorial effort is necessary.

Title, the use of ‘microtechnology’ is a bit confusing, perhaps use ‘tracking devices’? Also, I suggest to leave out the ‘Implications …. availability’ from the title since there were no measurements done of EA.

L17, in this context ‘stride’ is not clear.

Keywords, use capitals consistently.

L46-50, perhaps make the link to the recent observations by Ponzer et al. on metabolic adaptation.

L70-71, this sentence is hard to understand.

L80, ‘usage’ of persons? Does sound odd.

Figure 1, it took me some effort to understand the labels of the different types of activity. The reason was the way they are numbered, which I find confusing. I suggest to label the various exercises with letters, followed by the distance: A: 50m, B: 14m, etc. This will help the reader to get it right immediately.

L125-126, what were the magnetometers for?

L143, the editing post hoc should be explained more in detail.

L146, throughout the use of VO2 should be uniform. It now is a mix of upper, lower, subscript …

L146, is this system a mask based system?

L158, what is meant by ‘restructured’?

L203-207, hard to follow this section, please clarify.

Table 1, I would like to see the VO2 and actual distance data here too.

L232, typo in the GPS.

L233, spell out the unit for VO2.

L254, ‘demonstrate’? Perhaps use ‘verify’? ‘Investigate’? ‘Quantify’?

L274, which authors are meant here?

L276, not clear what is discussed here.

L301, what is meant with this tendency?

Author Response

Please see the attachment for in-depth answer to the comments made 

Round 2

Reviewer 3 Report

I thank the authors for taking into account my criticisms. I am satisfied with the answers and the changes made. A list editorial effort is necessary, there are some typos left.